# GRADIENT-FREE PROXY FOR EFFICIENT LANGUAGE MODEL SEARCH

## ABSTRACT

The demand for efficient natural language processing (NLP) systems has led to the development of lightweight language models. Previous work in this area has primarily focused on manual design or training-based neural architecture search (NAS) methods. Recently, zero-shot NAS methods have been proposed for evaluating language models without the need for training. However, prevailing approaches to zero-shot NAS often face challenges such as biased evaluation metrics and computational inefficiencies. In this paper, we introduce weight-weighted PCA (W-PCA), a novel zero-shot NAS method specifically tailored for lightweight language models. Our approach utilizes two evaluation proxies: the parameter count and principal component analysis (PCA) value of the feed-forward neural (FFN) layer. This provides a comprehensive and unbiased assessment of the language model's performance. Additionally, by eliminating the need for gradient computations, we optimize the evaluation time, thus enhancing the efficiency of designing and evaluating lightweight language models. We conduct a comparative analysis on the GLUE and SQuAD datasets to evaluate our approach. The results demonstrate that our method significantly reduces training time compared to one-shot NAS methods and achieves higher scores in the testing phase compared to previous state-of-the-art training-based methods. Furthermore, we perform ranking evaluations on a dataset sampled from the FlexiBERT search space. Our approach exhibits superior ranking correlation and further reduces solving time compared to other zero-shot NAS methods that require gradient computation.

## 1 INTRODUCTION

Large language models (LLMs) have shown exceptional performance across various domains (OpenAI, 2023). However, their size and computational demands pose challenges in resource-constrained environments like mobile devices and edge computing. Therefore, there is a growing need to explore lightweight language models that can operate efficiently on these platforms. One approach to address this challenge is through knowledge distillation (KD) (Hinton et al., 2015), where a larger language model acts as a teacher to train a smaller, more lightweight language model (Turc et al., 2019; Sanh et al., 2020; Jiao et al., 2020; Sun et al., 2020; Wang et al., 2020). However, the student models trained for these tasks were manually designed. To effectively search for student models, the use of neural architecture search (NAS) has become essential.

NAS is a technique that automates the process of designing neural networks, enabling the exploration of a wide range of architectures to identify the most optimal ones for a given task. Vanilla NAS approaches primarily used reinforcement learning (Zoph & Le, 2016) or genetic algorithms (Real et al., 2019) to train neural networks from scratch, but these methods were computationally expensive.

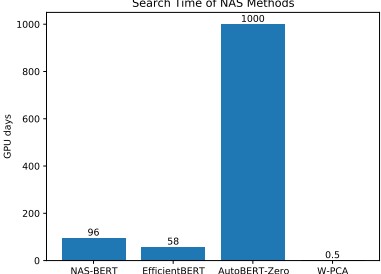

Figure 1: Comparison of the running time between W-PCA and other training-based NAS methods for lightweight language models. Our method achieves a substantial reduction in search time for the optimal network structure by two to three orders of magnitude, as we do not need to train the supernet.

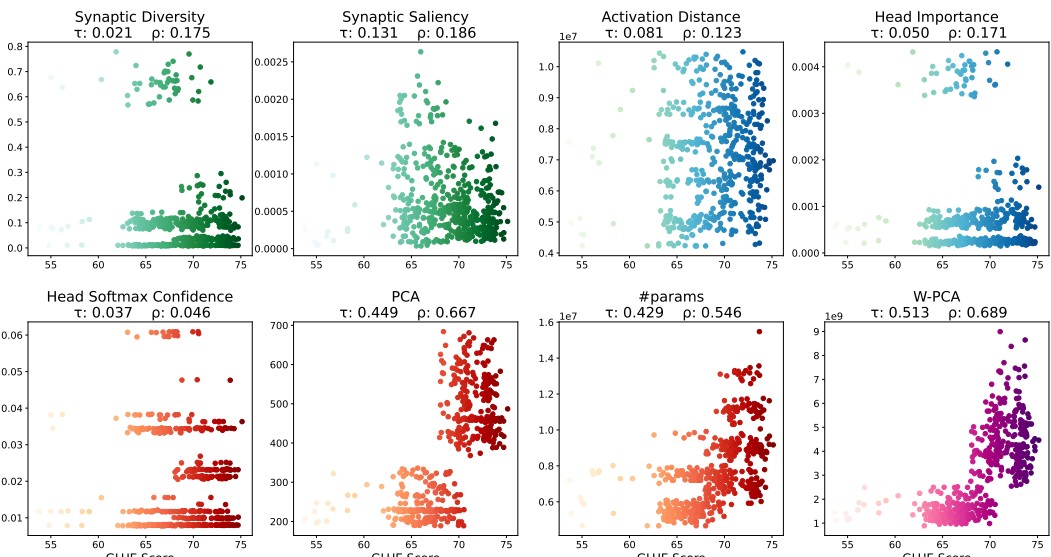

Figure 2: Plots depicting the evaluation of zero-shot proxy metrics on 500 randomly sampled architectures from the FlexiBERT search space, in relation to the GLUE score of the pretrained and finetuned architecture. The metric W-PCA is calculated as the product of the number of parameters (#params) and the principal component analysis (PCA).

Subsequently, one-shot NAS methods, such as gradient-based (Liu et al., 2018) and single path one-shot (SPOS) methods (Guo et al., 2020), were proposed. These methods are more efficient as they leverage pre-trained models or parameter sharing. Importantly, many lightweight model search tasks in natural language understanding (NLU) are accomplished using one-shot NAS (Xu et al., 2021; Dong et al., 2021; Gao et al., 2022). While one-shot NAS reduces training costs compared to training from scratch, it still requires various training strategies to effectively train the supernet. However, to further enhance search efficiency, it is necessary to introduce zero-shot NAS (Mellor et al., 2021). Zero-shot, also known as training-free NAS, is a promising approach that eliminates the need for training neural networks and directly evaluates their performance using proxy metrics. This significantly reduces the training time and computational resources required for NAS.

Existing zero-shot NAS methods (Abdelfattah et al., 2020; Zhou et al., 2022; Mellor et al., 2021; Celotti et al., 2020; Serianni & Kalita, 2023) have primarily focused on ranking correlations on NAS benchmark datasets (Klyuchnikov et al., 2022), with limited consideration for specific deep learning tasks. This limitation hinders their applicability and effectiveness in practical scenarios. Additionally, these methods often solely consider a single feature of the models, leading to biased evaluations and potentially overlooking important characteristics.

In our research, we aim to address these limitations and improve the applicability of zero-shot NAS. As shown in Figure 2, we attempted to incorporate previous zero-shot proxies into language model evaluation but obtained unsatisfactory results. However, we observed a strong correlation in ranking between principal component analysis (PCA) and the number of parameters (#params), with their product demonstrating even better performance.

Motivated by these findings, we propose a novel approach called Weight-Weighted PCA (W-PCA), which takes into account both the parameter count and PCA values of the model. By integrating these two factors, our aim is to achieve a more accurate and unbiased evaluation of language models in the context of zero-shot NAS. Furthermore, we have designed a search space specifically for NLU tasks and applied our designed zero-shot proxies, as well as the previous zero-shot proxies used in Transformer, to this search space. To the best of our knowledge, this is the first work that applies zero-shot NAS to NLU tasks.

## 2 RELATED WORK

### 2.1 LIGHTWEIGHT BERT MODELS

Turc et al. observed that distillation and pre-training + fine-tuning have mutually reinforcing effects. DistilBERT (Sanh et al., 2020) utilizes a triple loss function for training the lightweight model. TinyBERT (Jiao et al., 2020) applies distillation in both the pre-training and task-specific learning phases. MobileBERT (Sun et al., 2020) proposes a bottleneck structure to reduce the parameter count. MiniLM (Wang et al., 2020) introduces a compression method called deep self-attentive distillation. In this study, we incorporate both the standard BERT-base (Devlin et al., 2019) and MobileBERT models, along with their weight-sharing variations, where each layer is integrated into the supernet.

### 2.2 ONE-SHOT NAS FOR EFFICIENT MODELS

Numerous methods have been proposed for performing neural architecture search (NAS) to develop efficient models. NAS-BERT (Xu et al., 2021) trains a large supernet on a carefully designed search space that includes diverse architectures, generating multiple compressed models with adaptable sizes and latency. EfficientBERT (Dong et al., 2021) proposes a three-stage coarse-to-fine search scheme to optimize the combination of the multilayer perceptron (MLP) in the feed-forward network (FFN), ultimately reducing the parameter count of the FFN. AutoBERT-Zero (Gao et al., 2022) devises a search space that includes unary and binary math operators for constructing attention structures and backbones for general pre-trained language models (PLMs) from scratch. To the best of our knowledge, it is the most recent NAS method that incorporates lightweight BERT models in the experiment.

### 2.3 ZERO-SHOT NAS

Zero-shot NAS has been applied to transformer-based architectures in several ways. We provide a summary of these applications below.

**Synaptic Saliency** (Tanaka et al., 2020) aims to prevent layer collapse during network pruning, as this collapse can significantly reduce the accuracy of the network. The formulation for this approach is expressed as follows:

$$S(\theta) = \frac{\partial \mathcal{L}}{\partial \theta} \odot \theta$$

where $\mathcal{L}$ represents the loss function, $\theta$ denotes the network's parameters, and $\odot$ is the Hadamard product. Abdelfattah et al. generalize synaptic saliency as a zero-shot metric for NAS by summing over all $n$ parameters in the network: $S = \sum_{i=1}^{n} S(\theta_i)$

**Synaptic Diversity** builds upon previous research on rank collapse in transformers. In this phenomenon, the output of a multihead attention block tends to converge to rank 1 for a given set of inputs, which significantly impairs the performance of the transformer. Zhou et al. propose a method that utilizes the nuclear norm of an attention head's weight matrix $W_m$ as an approximation of its rank. This approach leads to the computation of the synaptic diversity score as follows:

$$S_D = \sum_m \| \frac{\partial \mathcal{L}}{\partial W_m} \|_{nuc} \odot \|W_m\|_{nuc}$$

**Activation Distance** is a proxy metric introduced by Mellor et al. to assess the ReLU activations of a network. By computing the Hamming distance between the activations within the initialized network for each input in a minibatch, this metric determines the similarity of the activation maps. The authors observe that when the activation maps for a given set of inputs exhibit higher similarity, the network faces greater difficulty in disentangling the input representations during the training process.

**Jacobian Covariance** evaluates the Jacobian $J = \left( \frac{\partial L}{\partial \mathbf{x}_1}, \ldots, \frac{\partial L}{\partial \mathbf{x}_N} \right)$ of the network's loss function with respect to the minibatch inputs. Further details of this metric can be found in the original paper (Mellor et al., 2021).

**Jacobian Cosine** (Celotti et al., 2020) is proposed as an improvement to the Jacobian Covariance metric, aiming to enhance computation speed and effectiveness. This improvement involves utilizing cosine similarity instead of a covariance matrix to measure similarity. The metric is computed as follows:

$$S = 1 - \frac{1}{N^2 - N} \sum_{i=1}^{N} |J_n J_n^T - I|^{\frac{1}{20}}$$

Here, $J_n$ represents the normalized Jacobian, and $I$ is the identity matrix. The metric is computed using a minibatch of $N$ inputs. In their large noise and more noised scores, the authors introduce various noise levels to the input minibatch, hypothesizing that architectures exhibiting high accuracy will demonstrate robustness against noise.

**Attention Confidence, Importance, and Softmax Confidence** "Confident" attention heads exhibit high attention towards a single token, indicating their potential importance to the transformer's task. Researchers have proposed different approaches to calculating confidence, including examining the softmax layer of the attention head and analyzing the sensitivity of the attention head to weight masking by computing the product between the attention head's output and the gradient of its weights. Serianni & Kalita summarize the findings from (Voita et al., 2019; Behnke & Heafield, 2020; Michel et al., 2019) regarding the following metrics:

Confidence: $A_h(\mathbf{X}) = \frac{1}{N} \sum_{n=1}^{N} |\max(Att_h(\mathbf{x}_n))|$

Softmax Confidence: $A_h(\mathbf{X}) = \frac{1}{N} \sum_{n=1}^{N} |\max(\sigma_h(\mathbf{x}_n))|$

Importance: $A_h(\mathbf{X}) = |Att_h(\mathbf{X}) \frac{\partial \mathcal{L}(\mathbf{X})}{\partial Att_h(\mathbf{X})}|$

where $X = \{x_n\}_{n=1}^{N}$ represents a minibatch of $N$ inputs, $\mathcal{L}$ denotes the loss function of the model, and $Att_h$ and $\sigma_h$ denote an attention head and its softmax, respectively. To obtain an overall metric for the entire network, Serianni & Kalita extend these scores by averaging them across all $H$ attention heads: $\mathcal{A}(\mathbf{X}) = \sum_{h=1}^{H} \frac{1}{H} Att_h(\mathbf{X})$

## 3  OUR GRADIENT-FREE WEIGHT-WEIGHTED PCA PROXY

**Vanilla PCA Proxy.** To evaluate the performance of candidate architectures, we utilize PCA (Principal Component Analysis) values of the BERT blocks. These values provide a measure of the valuable information contained in each dimension of the hidden states. Our metric is computed as follows:

$$S_f(\mathbf{X}) = \frac{1}{N} \sum_{i=1}^{N} \text{PCA\_dim}(\mathbf{x_i}, \eta) \tag{1}$$

where $\mathbf{X} = \{\mathbf{x_i}\}_{i=1}^{N}$ represents a minibatch of $N$ inputs, where "dim" refers to the dimension. We calculate the PCA values for the hidden states after the initial linear transformation in the FFN layer. By analyzing PCA_dim (the dimensions with PCA values exceeding a threshold $\eta$), we can identify the dimensions that contain a higher amount of valuable information. The metric for an $m$-layer neural network model is obtained by summing $S_f(\mathbf{X})$ over all layers, resulting in:

$$S(\mathbf{X}) = \sum_{f=1}^{m} S_f(\mathbf{X}) \tag{2}$$

where $\text{PCA\_dim}(\mathbf{x_i}, \eta)$ refers to the PCA value of the vector $\mathbf{x_i}$ in a specific dimension, considering a threshold value $\eta$. The metric $S_f(\mathbf{X})$ represents the PCA-based value for a specific layer $f$.

In order to calculate the PCA values, we first need to compute the covariance matrix. Let's dive into the details of how the covariance matrix is computed. Given a minibatch of $N$ hidden states represented by the matrix $\mathbf{X} = \{\mathbf{x_i}\}_{i=1}^{N}$, where each $\mathbf{x_i}$ is a hidden state vector, we want to calculate the covariance matrix $\mathbf{C}$. The covariance matrix $\mathbf{C}$ is an $D \times D$ symmetric matrix, where $D$ represents the dimensionality of the hidden states. The $(i, j)$-th element of the covariance matrix $\mathbf{C}$, denoted as

$C_{ij}$, captures the covariance between the $i$-th dimension and the $j$-th dimension of the hidden states. To compute $C_{ij}$, we need to calculate the covariance between the $i$-th and $j$-th dimensions across the minibatch. The covariance between two dimensions is a measure of how those dimensions vary together. The formula to compute the covariance between the $i$-th and $j$-th dimensions is as follows:

$$C_{ij} = \frac{1}{N} \sum_{k=1}^{N} (x_{ik} - \bar{x}_i)(x_{jk} - \bar{x}_j) \tag{3}$$

Here, $x_{ik}$ and $x_{jk}$ represent the $i$-th and $j$-th components of the hidden state vectors $\mathbf{x_k}$, respectively. $\bar{x}_i$ and $\bar{x}_j$ represent the mean values of the $i$-th and $j$-th dimensions across the minibatch, respectively. In other words, we compute the covariance matrix by taking the average of the pairwise products of the centered values of the hidden state dimensions. Once we compute all the elements of the covariance matrix following the above formula, we obtain the full covariance matrix $\mathbf{C}$. The covariance matrix $\mathbf{C}$ provides information about how the different dimensions of the hidden states vary together across the minibatch. This matrix is then used in the subsequent steps of the methodology, such as eigendecomposition to obtain the principal components and their corresponding eigenvalues.

Next, we perform eigendecomposition on the covariance matrix $\mathbf{C}$, which yields its eigenvectors and eigenvalues. The eigenvectors form the principal components, while the eigenvalues represent the importance of the corresponding eigenvectors. We sort the eigenvectors in descending order of their eigenvalues. By examining the dimensions with PCA values surpassing the threshold $\eta$, we can identify the dimensions that contain a higher amount of valuable information. These dimensions are considered important for evaluating the performance of the candidate architectures.

Finally, to compute the overall metric $S(\mathbf{X})$ for an $m$-layer neural network model, we sum the layer-specific metrics $S_f(\mathbf{X})$ over all layers. By utilizing this methodology, we can effectively assess the performance of candidate architectures based on their PCA values and identify the dimensions that contribute significantly to the valuable information in the hidden states.

**Weight-weighted PCA Proxy.** Besides PCA, we also observe that the weight parameters have strong correlation and are easy to compute. Thus, we propose a new metric called W-PCA, which combines the number of parameters (denoted as $w$) with the PCA values to assess the performance of candidate architectures. The W-PCA metric quantifies the amount of valuable information captured by each dimension relative to the number of parameters in the architecture.

The W-PCA metric is computed as the product of the number of weight parameters ($w$) and the PCA value for each dimension. Mathematically, it can be expressed as:

$$\text{W-PCA}(\mathbf{X}) = w \times S(\mathbf{X}) \tag{4}$$

The W-PCA metric provides a comprehensive evaluation of candidate architectures by considering both their complexity (number of parameters) and the amount of valuable information present in each dimension. By incorporating PCA, we can identify dimensions that contribute significantly to the overall performance of the architecture.

Advantages of our method include:

1. **Strong Correlation:** The W-PCA metric captures the relationship between the number of parameters and the valuable information in each dimension. This relevance is crucial in evaluating the efficiency and effectiveness of candidate architectures. By considering the PCA values, we can identify dimensions that contribute the most to the architecture's performance, allowing for informed decision-making during architecture search.

2. **Gradient-Free:** Unlike many traditional optimization methods that rely on gradients, our methodology is gradient-free. This eliminates the need for extensive backpropagation and derivative calculations, making the evaluation process more efficient and less computationally expensive.

3. **One forward propagation only:** Our methodology requires only forward propagation during the evaluation of candidate architectures. This simplifies the implementation and reduces the computational overhead, as it avoids the need for complex and resource-intensive operations such as backpropagation.

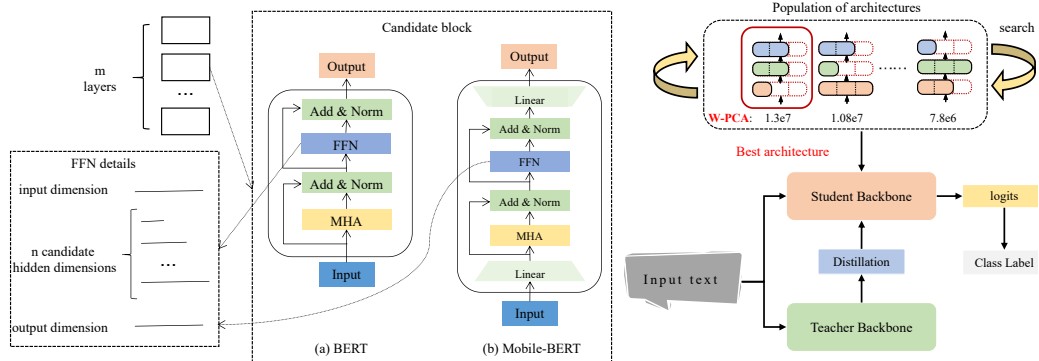

Figure 3: Overview of the W-PCA framework for NLU tasks. We employ a genetic algorithm to identify the optimal structure with the highest W-PCA value from the combinations in $(2 \times n)^m$. Subsequently, we refine it through additional training using knowledge distillation (KD). FFN and MHA represent the feed-forward network and multi-head attention, respectively.

By leveraging the advantages of strong relevance, gradient-freeness, and the use of only forward propagation, our methodology based on the W-PCA metric provides an efficient and effective approach for training-free architecture search. It enables researchers and practitioners to evaluate candidate architectures based on their valuable information content relative to the number of parameters, facilitating the exploration of architecture design space and aiding in the development of more efficient and effective models.

## 4 SEARCH SPACE FOR NLU TASKS

To enhance the evaluation of W-PCA's performance on NLU tasks, we have meticulously crafted a search space. Drawing inspiration from SPOS (Guo et al., 2020), our search targets a model comprised of multiple layers, with each layer capable of being a lightweight BERT model. The hidden dimension of the FFN layer within each BERT block is determined through random selection. This deliberate randomness aids in exploring a diverse range of architectures during the search process.

## 5 RANKING EVALUATION

### 5.1 DATASETS

Table 1: Comparison of different zero-shot proxies on the FlexiBERT benchmark. "Time" represents the computation time for the metric calculated 1,000 times.

| Proxy | Time | ∇-free | $\tau$ | $\rho$ |
|---|---|---|---|---|
| Synaptic Diversity (Zhou et al., 2022) | 110 s | ✗ | 0.021 | 0.175 |
| Synaptic Saliency (Abdelfattah et al., 2020) | 121 s | ✗ | 0.157 | 0.266 |
| Activation Distance (Mellor et al., 2021) | 68 s | ✓ | 0.081 | 0.123 |
| Jacobian Cosine (Celotti et al., 2020) | 103 s | ✗ | 0.116 | 0.149 |
| Head Importance (Serianni & Kalita, 2023) | 112 s | ✗ | 0.050 | 0.171 |
| Head Confidence (Serianni & Kalita, 2023) | 81 s | ✓ | 0.306 | 0.364 |
| Vanilla PCA | 61 s | ✓ | 0.449 | 0.667 |
| W-PCA | 74 s | ✓ | **0.513** | **0.689** |

To assess the accuracy of the proposed proxy indicators for neural network evaluation, we employed a benchmark consisting of a well-trained BERT structure suggested by Serianni & Kalita as the testing dataset. Specifically, this benchmark selected 500 structures from the FlexiBERT (Tuli et al., 2023) search space (as presented in Table 6) and utilized ELECTRA (Clark et al., 2020), rather than the MLM method, for training to efficiently pretrain a compact BERT model. The training dataset comprised 8,013,769 documents sourced from the OpenWebText (Gokaslan et al., 2019) corpus, amounting to a total of 38GB. For detailed training information, please refer to Appendix A. After training, the scores obtained by fine-tuning on the GLUE dataset will serve as the reference for evaluating the correlation among different zero-shot proxies.

## 5.2 RESULTS AND ANALYSIS

The evaluation results comprise the Kendall rank correlation coefficient (Kendall $\tau$) and the Spearman rank correlation coefficient (Spearman $\rho$). As shown in Table 1, Vanilla PCA has already exceeded the previous zero-shot proxy in terms of ranking correlation, and W-PCA performs even better than Vanilla PCA. Moreover, the absence of gradient computation further boosts the computational efficiency of W-PCA compared to proxies.

## 6 ACCURACY COMPARISION

### 6.1 DATASETS

To enable an accurate comparison to other lightweight BERT models, we evaluate the performance of W-PCA using the GLUE (Wang et al., 2018) and SQuAD (Rajpurkar et al., 2016) datasets and their corresponding task-specific evaluations.

### 6.2 IMPLEMENTATION DETAILS

#### 6.2.1 SEARCH SPACE

The search space is shown in Figure 3, where we set the values of $m$ set to 12 and $n$ set to 6, respectively. Each block has a hidden size of 528, with the inner hidden size of the MobileBERT series blocks being one-fourth of the total hidden size. The hidden dimensions of the FFN increase by a factor of 132 for each multiple from 1 to $n$. The value of PCA_dim is calculated with $\eta$ set to 0.99. We employ a genetic algorithm with a population size of 50 and a generation count of 40 to identify the combination of blocks that yields the highest PCA value. The crossover probability is set to 1, the mutation probability to 0.1, and the upper limit for the model parameters to 15.7M in order to obtain the W-PCA-Small model. By further reducing the upper limit for the model parameters to 10M and halving the number of layers ($m$), we obtain the W-PCA-Tiny model.

#### 6.2.2 TRAINING

Once we obtain the desired architecture, we pretrain the model using the complete English Wikipedia (Devlin et al., 2019) and BooksCorpus (Zhu et al., 2015). We then proceed to fine-tune the model on each individual downstream task. During pretraining, the network is trained with a batch size set to 256. For the fine-tuning phase of the downstream tasks, the network is trained with a batch size set to 32. The CoLA task is trained for 50 epochs, while the other tasks are trained for 10 epochs. The learning rate is set at 0.0001 during pretraining. In the fine-tuning phase, the learning rate is set at 0.00005 for GLUE tasks and 0.0001 for SQuAD tasks. The training process utilizes the Adam optimizer with $\beta_1$ and $\beta_2$ values set at 0.9 and 0.999, respectively. The weight decay is set to 0.01. The learning rate decays linearly with a warm-up ratio set to 0.1. The KD loss function used in our approach is described in Appendix C.

### 6.3 RESULTS ON GLUE

#### 6.3.1 MODEL ACCURACY AND LATENCY

Table 2 presents the results of the GLUE scores and model latency for the KD-based methods. Among them, except for the BERT-base teacher model we used ourselves, the results of all the manual and one-shot methods in the table are from relevant papers. Since zero-shot NAS methods have not been used in NLU tasks before, we applied the recent top-performing zero-shot proxy approaches on Transformer language models to the search space shown in Figure 3.

As shown in Table 2, under the search space depicted in Figure 3, our W-PCA metric achieved higher average scores on the GLUE test set compared to all baseline manual and one-shot methods. At the same time, it outperformed the previous state-of-the-art (SOTA) method EfficientBERT (Dong et al., 2021) in terms of parameter count, latency, and average score in the field of lightweight models. Additionally, W-PCA achieved the highest score on the STS-B task. It is worth noting that, in the same search space, the optimal structure found by W-PCA surpasses all previous zero-shot methods

Table 2: Performance comparison of the test set on the GLUE benchmark. The performance of all zero-shot proxies is evaluated on the search space depicted in Figure 3. Latency measurements of the models are conducted using the NVIDIA A100 GPU.

| Model | Type | #Params | Latency | QNLI | MRPC | SST-2 | CoLA | STS-B | MNLI-m/mm | RTE | QQP | AVG |
|---|---|---|---|---|---|---|---|---|---|---|---|---|
| BERT-base (Devlin et al., 2019) | manual | 108.9M | 274ms | 90.5 | 88.9 | 93.5 | 52.1 | 85.8 | 84.6/83.4 | 66.4 | 71.2 | 79.6 |
| BERT-base (ours) | manual | 108.9M | 274ms | 91.4 | 88.7 | 93.0 | 49.0 | 87.5 | 84.9/83.9 | 76.6 | 71.3 | 80.7 |
| BERT-tiny (Turc et al., 2019) | manual | 14.5M | 44ms | 84.8 | 83.2 | 87.6 | 19.5 | 77.1 | 75.4/74.9 | 62.6 | 66.5 | 70.2 |
| BERT-small (Turc et al., 2019) | manual | 28.8M | 79ms | 86.4 | 83.4 | 89.7 | 27.8 | 77.0 | 77.6/77.0 | 61.8 | 68.1 | 72.1 |
| DistilBERT-6 (Sanh et al., 2020) | manual | 67.0M | 151ms | 88.9 | 86.9 | **92.5** | **49.0** | 81.3 | 82.6/81.3 | 58.4 | 70.1 | 76.8 |
| TinyBERT-4 (Jiao et al., 2020) | manual | 14.5M | 45ms | 87.7 | 88.5 | 91.2 | 27.2 | 83.0 | 81.8/80.7 | 64.9 | 69.6 | 75.0 |
| MobileBERT-tiny (Sun et al., 2020) | manual | 15.1M | 62ms | 89.5 | 87.9 | 91.7 | 46.7 | 80.1 | 81.5/81.6 | 65.1 | 68.9 | 77.0 |
| EfficientBERT+ (Dong et al., 2021) | one-shot | 15.7M | 62ms | 89.3 | **89.9** | 92.4 | 38.1 | 85.1 | **83.0**/82.3 | 69.4 | **71.2** | 77.9 |
| EfficientBERT++ (Dong et al., 2021) | one-shot | 16.0M | 65ms | **90.6** | 88.9 | 92.3 | 42.5 | 83.6 | **83.0**/82.5 | 67.8 | **71.2** | 78.0 |
| Synaptic Saliency (Abdelfattah et al., 2020) | zero-shot | 15.7M | 58ms | 89.4 | 88.1 | 91.0 | 33.6 | 83.1 | 82.6/81.1 | 70.6 | 70.3 | 76.6 |
| Activation Distance (Mellor et al., 2021) | zero-shot | 15.6M | 60ms | 88.9 | 87.6 | 91.2 | 30.7 | 82.9 | 81.1/80.4 | 70.4 | 70.1 | 75.9 |
| Synaptic Diversity (Zhou et al., 2022) | zero-shot | 15.6M | 57ms | 88.3 | 88.1 | 91.5 | 25.8 | 84.7 | 81.3/80.2 | 70.6 | 70.3 | 75.6 |
| Head Confidence (Serianni & Kalita, 2023) | zero-shot | 15.6M | 63ms | 89.5 | 88.3 | 92.4 | 31.7 | 85.7 | 82.8/81.9 | **74.0** | 70.9 | 77.5 |
| Softmax Confidence (Serianni & Kalita, 2023) | zero-shot | 15.6M | 61ms | 88.4 | 87.5 | 90.8 | 32.5 | 83.5 | 81.2/80.5 | 70.3 | 69.9 | 76.1 |
| W-PCA-Tiny | zero-shot | 9.6M | 38ms | 88.7 | 87.6 | 91.9 | 27.4 | 84.8 | 81.1/79.8 | 71.1 | 70.3 | 75.9 |
| W-PCA-Small | zero-shot | 15.6M | 54ms | 90.3 | 88.7 | 91.5 | 38.4 | **86.4** | 82.8/82.2 | 73.8 | 70.8 | **78.3** |

Table 3: Comparison of results on the GLUE dev set with other NAS methods. The "Time" column represents the GPU days consumed by the NAS method search. It is not feasible to make a sub-comparison with AutoBERT-Zero-small as it does not provide individual scores for each task in the GLUE dev set.

| Model | #Params | Time | QNLI | MRPC | SST-2 | CoLA | STS-B | MNLI-m | RTE | QQP | AVG |
|---|---|---|---|---|---|---|---|---|---|---|---|
| NAS-BERT-10 (Xu et al., 2021) | 10.0M | 96 d | 86.3 | 79.1 | 88.6 | 34.0 | 84.8 | 76.4 | 66.6 | 88.5 | 75.5 |
| NAS-BERT-30 (Xu et al., 2021) | 30.0M | 96 d | 88.4 | 84.6 | 90.5 | 48.7 | **87.6** | 81.0 | 71.8 | **90.2** | 80.3 |
| EfficientBERT-TINY (Dong et al., 2021) | 9.4M | 58 d | 89.3 | 90.1 | 90.1 | 39.1 | 79.9 | 81.7 | 63.2 | 86.7 | 77.5 |
| EfficientBERT (Dong et al., 2021) | 15.7M | 58 d | 90.4 | **91.5** | 91.3 | **50.2** | 82.5 | **83.1** | 66.8 | 87.3 | 80.4 |
| AutoBERT-Zero-small (Gao et al., 2022) | 13.0M | ˜1,000 d | - | - | - | - | - | - | - | - | 80.5 |
| Synaptic Diversity (Zhou et al., 2022) | 15.6M | 0.7 d | 88.9 | 87.6 | 91.4 | 32.0 | 84.1 | 81.0 | 73.4 | 88.2 | 78.3 |
| Head Confidence (Serianni & Kalita, 2023) | 15.6M | 0.5 d | 90.1 | 89.7 | 92.4 | 37.5 | 84.1 | 82.5 | 75.9 | 89.1 | 80.2 |
| Softmax Confidence (Serianni & Kalita, 2023) | 15.6M | 0.5 d | 89.4 | 88.3 | 92.0 | 32.6 | 84.7 | 81.6 | 73.9 | 88.9 | 78.9 |
| W-PCA-Tiny | 9.6M | 0.4 d | 89.2 | 89.2 | 92.0 | 33.2 | 84.0 | 80.5 | 71.1 | 88.0 | 78.4 |
| W-PCA-Small | 15.6M | 0.5 d | **90.8** | 90.5 | **92.8** | 44.0 | 85.3 | 82.9 | **76.1** | 88.8 | **81.4** |

(Abdelfattah et al., 2020; Mellor et al., 2021; Zhou et al., 2022; Serianni & Kalita, 2023) applied to Transformer language models, highlighting its exceptional ability in exploring optimal network structures in zero-shot NAS methods.

### 6.3.2 SEARCH EFFICIENCY

As shown in Table 3, under our search space, the search efficiency of all zero-shot proxies (including our W-PCA method) has been improved by two to three orders of magnitude compared to previous training-based NAS, and achieved competitive performance. The three zero-shot proxies, Synaptic Diversity (Zhou et al., 2022), Head Confidence (Serianni & Kalita, 2023), and Softmax Confidence (Serianni & Kalita, 2023), can compete with the optimal structures found by previous training-based NAS in our search space. Our W-PCA method surpasses all previous training-based methods in the field of lightweight language models in terms of average score and achieves the best average score. Moreover, in three out of eight tasks, W-PCA achieves the highest performance. Our method discovers the latest SOTA effects in the field of lightweight models with almost negligible search cost, reducing greenhouse gas $CO_2$ emissions by two to three orders of magnitude, and significantly improving the utilization of global energy resources.

It is also worth noting that in the internal comparison of zero-shot proxies, Head Confidence (Serianni & Kalita, 2023), Softmax Confidence (Serianni & Kalita, 2023), and our W-PCA method require shorter search time than the Synaptic Diversity (Zhou et al., 2022) method, which needs to compute gradients, by an additional 0.2 GPU days. Additionally, our W-PCA-Tiny model has a lower parameter limit set during the search, resulting in slightly lower computation time for the forward propagation of each neural network individual, thus reducing the search time by 0.1 GPU days compared to the W-PCA-Small model.

Table 5: Comparison results of W-PCA and its product counterparts as proxies on the GLUE dev set.

| Proxy | #Params | QNLI | MRPC | SST-2 | CoLA | STS-B | MNLI-m | RTE | QQP | AVG |
|---|---|---|---|---|---|---|---|---|---|---|
| #Params | 15.7M | 89.3 | 88.8 | 90.7 | 43.8 | 83.6 | 82.6 | 76.1 | 87.5 | 80.3 |
| V-PCA | 15.6M | 89.9 | 91.4 | 92.7 | 39.4 | 84.9 | 82.9 | 76.0 | 88.9 | 80.8 |
| W-PCA | 15.6M | 90.8 | 90.5 | 92.8 | 44.0 | 85.3 | 82.9 | 76.1 | 88.8 | 81.4 |

## 6.4 RESULTS ON SQUAD

We compared the W-PCA proposed in this article with manually designed lightweight models, namely TinyBERT (Jiao et al., 2020), MiniLM (Wang et al., 2020), and the one-shot NAS method EfficientBERT (Dong et al., 2021), on the SQuAD dataset. The results are presented in Table 4. Despite having fewer parameters than TinyBERT-4, MiniLM-6, and Efficient++, the W-PCA-Small model outperforms these methods in terms of both EM and F1 scores on both the SQuAD v1.1 and SQuAD v2.0 datasets. This observation demonstrates the robust adaptability of the investigated models across diverse datasets.

## 6.5 ABLATION STUDY

Table 4: Results on SQuAD dev sets. *: our implementation.

| Model | #Params | SQuAD v1.1 EM/F1 | SQuAD v2.0 EM/F1 |
|---|---|---|---|
| BERT-base | 108.9M | 80.8/88.5 | -/- |
| BERT-base* | 108.9M | 80.7/88.2 | 75.7/78.7 |
| TinyBERT-4 | 14.5M | 72.7/82.1 | 68.2/71.8 |
| MiniLM-6 | 22.9M | -/- | -/72.7 |
| EfficientBERT++ | 16.0M | 78.3/86.5 | 73.0/76.1 |
| W-PCA-Tiny | 9.6M | 74.6/83.5 | 69.0/72.1 |
| W-PCA-Small | 15.6M | **78.4/86.7** | **73.3/76.8** |

In order to investigate the effects of each component of W-PCA on the experimental results, we performed ablation experiments. Specifically, we utilized the components of W-PCA, as described in Equation (4) where the first component is the number of parameters (#Params) and the second component is the V-PCA value (defined in Equation (2)), as fitness values for the genetic algorithm to explore the optimal network structure in Section 6.2.1. We then compared the performance of the discovered network structures with W-PCA.

The results, presented in Table 5, demonstrate that by multiplying the number of parameters with the V-PCA value and using W-PCA as the zero-shot evaluation metric, the performance of the searched networks significantly improves compared to using either #Params or V-PCA alone as the evaluation metric.

Encouragingly, the inclusion of an additional feature does not require a substantial increase in computational time, making the multiplication approach highly efficient.

## 7 CONCLUSION

In this paper, we propose W-PCA, a novel zero-shot NAS method specifically designed for lightweight language models. In the ranking correlation experiments conducted on the search space of FlexiBERT, W-PCA achieves a Kendall $\tau$ score that surpasses the previous method by 0.207 and a Spearman $\rho$ score that surpasses the previous method by 0.325. In the accuracy experiments conducted on GLUE and SQuAD, W-PCA not only achieves the highest score, but also significantly improves search efficiency. On the GLUE test set, W-PCA improves search efficiency by over a hundredfold compared to the previous best-performing one-shot NAS method, with an average score improvement of 0.3. On the GLUE dev set, W-PCA improves search efficiency by 2,000 times and achieves an average score improvement of 0.9 compared to the previous best-performing one-shot NAS method. Our work contributes to the advancement of NAS methods for lightweight language models, enabling the design and optimization of efficient and effective systems for natural language processing.

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

## A  TRAINING DETAILS OF THE FLEXIBERT SEARCH SPACE

All transformer architectures within the search space were trained on TPUv2s with 8 cores and 64 GB of memory using Google Colaboratory. The entire process of pretraining and finetuning the benchmark took approximately 25 TPU days. For the evaluation of training-free metrics, 2.8 GHz Intel Cascade Lake processors with either 16 or 32 cores and 32 GB of memory were employed.

In terms of hyperparameter settings, except for setting the training steps to 100,000 during the pre-training phase, everything else is the same as training ELECTRA-Small. Specifically, during the pre-training phase, the generator size multiplier is set to 1/4, the mask percentage is set to 15%, the

Table 6: The FlexiBERT search space comprises a total of 10,621,440 architectures.

| Architecture Element | | Hyperparameters Values |
|---|---|---|
| Hidden dimension | | {128, 256} |
| Number of Encoder Layers | | {2, 4} |
| Type of attention operator | | {self-attention, linear transform, span-based dynamic convolution} |
| Number of operation heads | | {2, 4} |
| Feed-forward dimension | | {512, 1024} |
| Number of feed-forward stacks | | {1, 3} |
| Attention operation | if self-attention | {scaled dot-product, multiplicative} |
| | if linear transform | {discrete Fourier, discrete cosine} |
| | if dynamic convolution | convolution kernel size: {5, 9} |

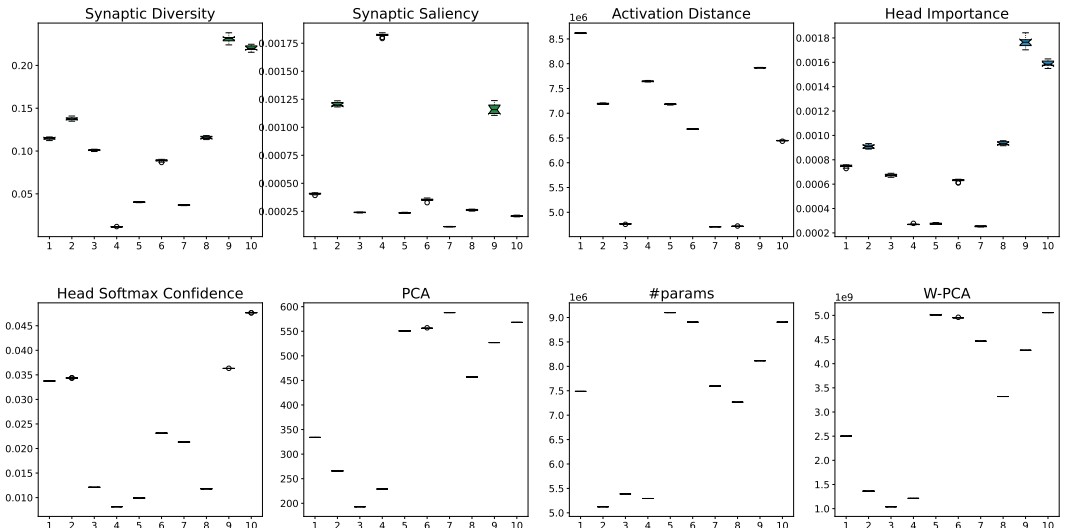

Figure 4: Evaluation of zero-shot metrics with various initialization weights in the FlexiBERT search space. A total of 10 architectures are randomly sampled from the search space, with each architecture representing a decile range of the GLUE score (e.g., 0-10%, 10-20%, ..., 90-100%). To ensure robustness, 10 different random seeds are employed for initializing the weights.

warmup step is 10,000, the learning rate is 5e-4, and the batch_size is 128. During the fine-tuning phase, the learning rate is 3e-4, the layerwise $lr$ decay is 0.8, the warmup fraction is 0.1, the attention dropout is 0.1, and the batch_size is 32. For the RTE and STS tasks, 10 epochs are trained, while for other tasks, 3 epochs are trained. Both during pre-training and fine-tuning phases, the learning rate decay is linear, the vocabulary size is 30522, the dropout is 0.1, the weight decay value is 0.01, the $\epsilon$ value for the Adam optimizer is 1e-6, $\beta_1$ value is 0.9, and $\beta_2$ value is 0.999.

## B DISCUSSION OF DIFFERENT INITIALIZATION PARAMETERS

We conducted a series of studies to investigate the impact of random initialization of architectures on the evaluation of zero-shot metrics in the FlexiBERT search space. From Figure 4, it is evident that while the parameter quantity remains unaffected by weight initialization, the other zero-shot proxies exhibit varying degrees of fluctuations during the ten different weight initialization processes. However, our proposed method, W-PCA, demonstrates smaller fluctuation magnitudes and more stable performance compared to the proxies of Synaptic Diversity (Zhou et al., 2022), Synaptic Saliency (Abdelfattah et al., 2020), and Head Importance (Serianni & Kalita, 2023), which exhibit larger fluctuations.

## C  KD LOSS FUNCTION

The distillation loss function of EfficientBERT (Dong et al., 2021) forms the basis of our approach. For the student model, we define $\mathcal{L}_{attn}^i$ as the loss for the multi-head attention (MHA) output and $\mathcal{L}_{hidd}^i$ as the loss for the feed-forward network (FFN) output in the $m$-th layer. The embedding loss, represented by $\mathcal{L}_{embd}$, is also included. These losses are calculated using the mean squared error (MSE) as follows:

$$\begin{cases} \mathcal{L}_{attn}^i = \mathrm{MSE}(\mathbf{A}_i^S \mathbf{W}_a, \mathbf{A}_j^T), \\ \mathcal{L}_{hidd}^i = \mathrm{MSE}(\mathbf{H}_i^S \mathbf{W}_h, \mathbf{H}_j^T), \\ \mathcal{L}_{embd} = \mathrm{MSE}(\mathbf{E}^S \mathbf{W}_e, \mathbf{E}^T) \end{cases} \tag{5}$$

Here, $\mathbf{A}_i^S$ and $\mathbf{H}_i^S$ represent the outputs of the MHA and FFN layers, respectively, in the $i$-th layer of the student model. Similarly, $\mathbf{A}_j^T$ and $\mathbf{H}_j^T$ represent the outputs of the MHA and FFN layers, respectively, in the $j$-th layer of the teacher model corresponding to the $i$-th layer of the student model.

For our fixed teacher model, BERT-base, which comprises 12 layers, a one-to-one sequential correspondence exists between the layers of the student and teacher models when both models have 12 layers. However, in the case of a student model with only 6 layers, the correspondence remains one-to-one, but with a 2-layer interval. This implies that the first layer of the student model corresponds to the second layer of the teacher model, and so forth, until the sixth layer of the student model aligns with the twelfth layer of the teacher model.

The trainable matrices $\mathbf{W}_a$, $\mathbf{W}_h$, and $\mathbf{W}_e$ are used to adjust the dimensionality of the student and teacher models. Additionally, we define $\mathcal{L}_{pred}$ as the prediction loss, which is calculated using soft cross-entropy (CE):

$$\mathcal{L}_{pred} = \mathrm{CE}(\mathbf{z}^S, \mathbf{z}^T) \tag{6}$$

Here, $\mathbf{z}$ represents the predicted logit vector.

The total loss is a combination of the above terms:

$$\mathcal{L} = \sum_{i=1}^{m} (\mathcal{L}_{attn}^i + \mathcal{L}_{hidd}^i) + \mathcal{L}_{embd} + \gamma \mathcal{L}_{pred} \tag{7}$$

The coefficient $\gamma$ is used to control the contribution of the predicted loss. It is set at 0 during the pretraining phase and 1 during the fine-tuning phase.

## D  VISUALIZATION OF ARCHITECTURES

Figure 5 illustrates the schematic diagram of the network structure. It is observed that all models preferentially choose MobileBERT as the candidate block, suggesting that MobileBERT is better suited for lightweight language models in comparison to BERT-base. Furthermore, with the exception of the searched model that solely relies on parameter count as the search evaluation metric, the candidate blocks of MobileBERT are predominantly located in the higher layers, indicating that this architecture is more adept at analyzing high-level semantic information.

## E  FURTHER DISCUSSION ON NLU TASKS

### E.1  COMPARISON OF LARGER-SIZED MODELS

We conducted experiments using a larger-sized model by increasing the size of the search space, as described in Section 6.2.1. Specifically, we doubled the hidden_size and the hidden_dimension of $n$ candidate dimensions. Additionally, we raised the parameter limit in the genetic algorithm to 67M, resulting in our W-PCA-Large model. As shown in Table 7, despite having a slightly lower parameter count, our model outperforms TinyBERT-6 (Jiao et al., 2020) and EfficientBERT

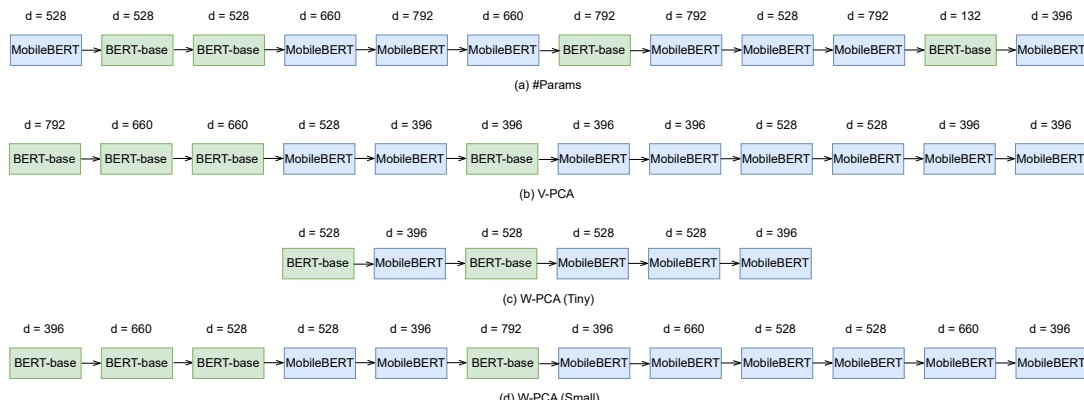

Figure 5: Visualizations of the searched architectures, where d represents the hidden dimensions.

Table 7: Performance comparison of larger-scale models on the GLUE test set.

| Model | #Params | QNLI | MRPC | SST-2 | CoLA | STS-B | MNLI-m/mm | RTE | QQP | AVG |
|---|---|---|---|---|---|---|---|---|---|---|
| TinyBERT-6 (Jiao et al., 2020) | 67.0M | 89.8 | 89.0 | 92.0 | 38.8 | 83.1 | 83.8/83.2 | 65.8 | 71.4 | 77.4 |
| EfficientBERT (Dong et al., 2021) | 70.1M | 90.4 | 89.0 | 92.6 | 46.2 | 83.7 | 84.1/83.2 | 67.7 | 74.4 | 78.7 |
| W-PCA-Large | 66.9M | 90.9 | 88.7 | 93.0 | 40.0 | 87.5 | 84.6/83.3 | 75.6 | 71.5 | 79.5 |

(Dong et al., 2021) models of similar scale in terms of average GLUE score. This indicates that our proposed zero-shot proxy also demonstrates good adaptability in larger search spaces.

### E.2 COMPARISON WITH ONE-SHOT NAS

**SPOS method.** We applied the one-shot NAS method in the same search space. Specifically, we constructed a supernet as shown in Figure 3, which consists of a total of $m$ layers. Each layer is composed of $2 \times n$ candidate blocks. Before searching for the optimal structure using a genetic algorithm, we performed one round of pre-training and fine-tuning based on the SPOS method (Guo et al., 2020). During each batch, a random path is selected from the $(2 \times n)^m$ combinations for forward propagation and backward parameter updates.

**Implementation details.** We first pretrain the supernet on English Wikipedia (Devlin et al., 2019) and BooksCorpus (Zhu et al., 2015), then utilize 90% of the training set from each GLUE task for fine-tuning. We reserve the remaining 10% of the MNLI task to evaluate the accuracy of architectures in the search. During the pre-training and fine-tuning process, the number of epochs is set to 10, and the batch_size is set to 256 for both. The learning rate for pre-training is set to 1e-4, and the learning rate for fine-tuning is set to 4e-4. The optimizer, weight decay, and learning rate adjustment strategy are the same as in the training section. The loss function used is still the MSE loss function described in Appendix C. After completing this pre-training and fine-tuning process, we proceed with the workflow described in the main text.

**Results & Analysis.** As shown in Table 8, despite investing a significant number of GPU days in the one-shot NAS search, the performance improvement on various-sized models of W-PCA is not significant. Zero-shot NAS remains the most cost-effective search solution.

Table 8: Comparison of zero-shot and one-shot methods on the GLUE test set in the same search space. "Time" also refers to the GPU time consumption in the NAS stage.

| Model | Type | #Params | Time | QNLI | MRPC | SST-2 | CoLA | STS-B | MNLI-m/mm | RTE | QQP | AVG |
|---|---|---|---|---|---|---|---|---|---|---|---|---|
| W-PCA-Tiny | zero-shot | 9.6M | 0.4 d | 88.7 | 87.6 | 91.9 | 27.4 | 84.8 | 81.1/79.8 | 71.1 | 70.3 | 75.9 |
| | one-shot | 9.7M | 24 d | 89.2 | 87.5 | 92.3 | 28.9 | 83.7 | 81.4/80.5 | 71.4 | 70.5 | 76.2 |
| W-PCA-Small | zero-shot | 15.6M | 0.5 d | 90.3 | 88.7 | 91.5 | 38.4 | 86.4 | 82.8/82.2 | 73.8 | 70.8 | 78.3 |
| | one-shot | 15.6M | 28 d | 90.3 | 88.9 | 92.5 | 36.1 | 86.7 | 83.7/82.5 | 74.4 | 70.6 | 78.4 |

