# OpenReview forum: "Gradient-free Proxy for Efficient Language Model Search"
_ICLR.cc/2024/Conference — Submitted to ICLR 2024_

### Official Review · Reviewer_7PxN · 2023-10-28

**Soundness:** 2 fair
**Presentation:** 2 fair
**Contribution:** 2 fair
**Rating:** 5
**Confidence:** 2

**Summary:**

This work introduces a novel zero-shot NAS method, called as weight-weighted PCA (W-PCA), designed to efficiently explore lightweight language models within a teacher-student network for knowledge distillation. The method harnesses the parameter counts and principal component analysis (PCA) values of the feed-forward neural (FFN) layer, instead of relying on gradient metrics, to offer a comprehensive and impartial evaluation. In experimental trials, this approach demonstrates the ability to significantly reduce training time when compared to previous one-shot NAS methods.

**Strengths:**

This work presents a zero-shot NAS method that:
1. Identifies significant dimensions that contribute to the performance.
2. Reduces the need for extensive backpropagation and derivative calculations and requires only forward propagation during the evaluation of candidate architectures.

**Weaknesses:**

1. The novelty may be limited, as the proposed method has not provided a comprehensive and robust architectural design guideline or led to the discovery of an efficient and universally applicable architecture.
2. Tables 2 and 3 demonstrate only modest performance improvements with W-PCA in comparison to baseline methods, with exceptions on a few datasets. This might raise questions about the overall effectiveness of the proposed method.

**Questions:**

1. “By considering the PCA values, we can identify dimensions that contribute the most to the architecture’s performance, allowing for informed decision-making during architecture search.” How will the dimensions benefit the architecture search?
2. Are there any particularities of the architectures searched by your method?

---

> ### Author Response · Authors · 2023-11-23
> **Response to Reviewer 7PxN**
>
> Thank you for your valuable feedback and constructive comments have been instrumental in shaping our research.
>
> > The novelty may be limited, as the proposed method has not provided a comprehensive and robust architectural design guideline or led to the discovery of an efficient and universally applicable architecture.
>
> We conducted additional experiments by applying the zero-shot proxy mentioned in the related work to our search space and discussed the impact of the search space on the experimental results. The experimental results have been added to Tables 2 and 3. Additionally, relevant discussions have been included in Section 6.3.
>
> > Tables 2 and 3 demonstrate only modest performance improvements with W-PCA in comparison to baseline methods, with exceptions on a few datasets. This might raise questions about the overall effectiveness of the proposed method.
>
> As mentioned above, we have supplemented the relevant experimental content in Tables 2 and 3.
>
> > “By considering the PCA values, we can identify dimensions that contribute the most to the architecture’s performance, allowing for informed decision-making during architecture search.” How will the dimensions benefit the architecture search?
>
> In general, the larger the dimension of PCA, the more effective information is contained in this block. In this regard, we are more or less inspired by the paper [1].
>
> > Are there any particularities of the architectures searched by your method?
>
> We have supplemented the discussion on model visualization in Appendix D.
>
> [1] Pan et al. Budgeted Training for Vision Transformer, ICLR 2023.

---

### Official Review · Reviewer_EqU6 · 2023-11-01

**Soundness:** 4 excellent
**Presentation:** 4 excellent
**Contribution:** 3 good
**Rating:** 8
**Confidence:** 4

**Summary:**

In this paper, the authors aim to address these limitations and improve the applicability of zero-shot NAS. As prevailing approaches to NAS often confront issues such as biased evaluation metrics and computational inefficiencies, the authors proposed w-PCA (given an observed strong correlation in ranking between PCA and the # of parameters (#params), with their product demonstrating even better performance) to better and more inclusively consider both the model parameters count and PCA values. Combining these two aspects, w-PCA shows significantly smaller training time compared to 1-shot NAS, while achieving higher scores in the testing phase compared to previous SOTA training-based methods.

**Strengths:**

1. The proposed method (w-PCA) achieved significantly smaller training time compared to 1-shot NAS
2. w-PCA achieving higher scores in the testing phase compared to previous SOTA training-based methods on GLUE and SQuAD datasets
3. Concluding experiments on two widely used NLU datasets & detailed analyses of the results are provided
4. First work that applies 0-shot NAS to NLU tasks
5. Details of the implementation (in the main text and appendix A), making comprehension easier, and reproducibility as well

**Weaknesses:**

1. The method struggled in performance on the GLUE & SQuAD datasets (e.g. BERT-base* performed the best, even if it is by a 2-point margin). Do you have any idea why?

2. Does the work scale? It would have been nice to discuss the limitations of this approach. For instance, I see this approach to be useful in resource constraint settings e.g. low-resource scenarios but the datasets used cover very few to none of such languages. It would be great to have it included to see to which extent this could benefit extremely constrained environments

3. What should be the trade-off between training time, the weight of the obtained model, and the performance? i.e. how much decrease (worse case) in performance could we "sacrifice" for how many # of parameters and how much time should/could?

4. Not related to the PCA necessarily but to the idea of reduction - have you tried or explored working in latent space?

**Questions:**

See Weaknesses

---

> ### Author Response · Authors · 2023-11-23
> **Response to Reviewer EqU6**
>
> Thank you for your valuable feedback and constructive comments have been instrumental in shaping our research.
>
> > The method struggled in performance on the GLUE & SQuAD datasets (e.g. BERT-base* performed the best, even if it is by a 2-point margin). Do you have any idea why?
>
> In fact, BERT-base is the teacher model, while our student model is based on knowledge distillation training, so the results generally do not surpass the teacher model.
>
> > Does the work scale? It would have been nice to discuss the limitations of this approach. For instance, I see this approach to be useful in resource constraint settings e.g. low-resource scenarios but the datasets used cover very few to none of such languages. It would be great to have it included to see to which extent this could benefit extremely constrained environments
>
> We have applied the previously used zero-shot proxy, which was originally applied to the Transformer block, in the search space we designed. The results have been added to Table 2 and Table 3, and relevant discussions have been included in Section 6.2. We have also added a discussion on parameter initialization in Appendix B.
>
> > What should be the trade-off between training time, the weight of the obtained model, and the performance? i.e. how much decrease (worse case) in performance could we "sacrifice" for how many # of parameters and how much time should/could?
>
> We compared our method with one-shot NAS, and the results are shown in Appendix E.2. In fact, spending more search time does not necessarily lead to significantly better results. Additionally, more constraints can be set as search conditions for future work.
>
> > Not related to the PCA necessarily but to the idea of reduction - have you tried or explored working in latent space?
>
> In fact, the discussion on KD loss in our current Appendix C should involve relevant categories, as the output of each layer can also be understood as a representation of a latent space.

---

### Official Review · Reviewer_pvZH · 2023-11-01

**Soundness:** 2 fair
**Presentation:** 2 fair
**Contribution:** 2 fair
**Rating:** 5
**Confidence:** 4

**Summary:**

This paper proposes a zero-shot neural architecture search method for selecting lightweight language models. The method involves leveraging two evaluation proxies - parameter dimension and eigen/spectral values. This methodology enables faster throughput of evaluating and selecting lightweight language models via gradient free computations. Evaluation on the GLUE benchmark shows higher scores as compared to previous state-of-the-art methods.

**Strengths:**

1. Zero-shot NAS enables faster searching of the models 0.5d as compared to > 50d using other approaches.

2. Competitive results against multiple baselines.

**Weaknesses:**

1. The paper is convoluted and difficult to understand. Multiple sections need to be written again. Kindly see Questions/Comments.

2. Table 5 results are close to each other. Thereby showing that the weight-weighted version might not improve the results too much as compared to Vanilla. A deeper qualitative analysis is required to establish the usage which is missing in the current version of this work.

**Questions:**

1. The first two lines of the abstract are not well connected. Kindly reframe the abstract to highlight the motivation and contribution better

2. Abstract: Unbiased assessment of what?

3. Section 1. 3rd last paragraph - “potentiallly overlooking important characteristics”. What are these important characteristics?

4. Section 2.2 mentions primitive operators - What are these?

5. Section 2.3: Include year with the citations.

6. Section 2.3: Inconsistent notations throughout this section - N is parameters, as well as batch size in this section. This section needs to be written better to make the reader understand the motivation for using each of the metrics, instead of just writing the formulation.

7. Section 3: Kindly mention how PCA_dim is calculated using equations, and how the threshold serves as the lower/upper limit for deciding the principal components. Explicitly call out hidden_dim = dimension of the hidden layer after before applying FFN.

8. Why is the scaling of hidden_dim needed ?

9. Equation 4: Notation writeup should be improved currently it looks like PCA(X) subtracted from W.

10. Section 5.1: Kindly include the details of the appendix section of Serianni & Kalita. Each paper should be a standalone read.

11. Section 6.2.1: Details of the genetic algorithm being used is missing here. Kindly include it here or the appendix.

12. Section 6.2.3: KD Loss - $\mathcal{L}^i_{attn} = \text{MSE}(\mathbb{A}^{S}*i\mathbb{W}a, \mathbb{A}^T*j).

13. Section 6.2.3: KD Loss - “jth teacher model layer corresponds to ith student model layer” - How do we get this correspondence?

14. Conclusion: Kindly mention some quantitative metrics here. In the current version it looks like a paraphrase of the abstract.

---

> ### Author Response · Authors · 2023-11-23
> **Response to Reviewer pvZH (1/2)**
>
> Thank you for your valuable feedback and constructive comments have been instrumental in shaping our research.
>
> > The first two lines of the abstract are not well connected. Kindly reframe the abstract to highlight the motivation and contribution better
>
> Thank you very much for your correction.  We have reframed the abstract to better highlight the motivation and contribution of our work.
>
> >  Abstract: Unbiased assessment of what?
>
> The term "unbiased assessment" refers to the comprehensive evaluation of the language model's performance using the parameter count and PCA value. We have clarified this in the modified abstract.
>
> > Section 1. 3rd last paragraph - “potentiallly overlooking important characteristics”. What are these important characteristics?
>
> These characteristics include some features of neural networks extracted in previous zero-shot NAS, such as gradients, parameter sizes, and parameter quantities discussed in Section 2.3.
>
> > Section 2.2 mentions primitive operators - What are these?
>
> These include unary (such as negation, softmax) and binary (such as addition, multiplication) mathematical operations, which we have explained in the original text.
>
> > Section 2.3: Include year with the citations.
>
> Thank you for the correction. We have made the necessary amendments to the relevant citations.
>
> > Section 2.3: Inconsistent notations throughout this section - N is parameters, as well as batch size in this section. This section needs to be written better to make the reader understand the motivation for using each of the metrics, instead of just writing the formulation
>
> Thank you for your correction. We have replaced "n" with "N" as the parameter and added a description of the relevant motivation in the corresponding zero-shot proxy.
>
> > Section 3: Kindly mention how PCA_dim is calculated using equations, and how the threshold serves as the lower/upper limit for deciding the principal components. Explicitly call out hidden_dim = dimension of the hidden layer after before applying FFN.
>
> PCA_dim refers to the dimensions with PCA values exceeding a threshold $\eta$. We have added relevant explanations below Equation (1). In Section 6.2.1, we mentioned that the value of
> $\eta$ is set to 0.99 in the experiments.
>
> > Why is the scaling of hidden_dim needed ?
>
> Thank you for helping us reconsider the role of hidden_dim. After repeated experiments, we found that removing this scaling would lead to better correlation in the dataset rankings. We have updated the relevant experimental results.
>
> > Equation 4: Notation writeup should be improved currently it looks like PCA(X) subtracted from W.
>
> We have updated the expression of the relevant formula, and now the "-" in the formula no longer looks like a minus sign.
>
> > Section 5.1: Kindly include the details of the appendix section of Serianni & Kalita. Each paper should be a standalone read.
>
> Thank you for your correction. We have added the relevant content to Appendix A.
>
> > Section 6.2.1: Details of the genetic algorithm being used is missing here. Kindly include it here or the appendix.
>
> We employ a genetic algorithm with a population size of 50 and a generation count of 40 to identify the combination of blocks that yields the highest PCA value. The crossover probability is set to 1, the mutation probability to 0.1, and the upper limit for the model parameters to 15.7M in order to obtain the W-PCA-Small model. By further reducing the upper limit for the model parameters to 10M and halving the number of layers ($m$), we obtain the W-PCA-Tiny model.
>
> This part of the content has been written in Section 6.2.1.
>
> > Section 6.2.3: KD Loss - $\mathcal{L}^i_{attn} = \text{MSE}(\mathbb{A}^{S}i\mathbb{W}a, \mathbb{A}^Tj).
>
> Thank you for your correction. We have made the necessary revisions. We have moved all the sections related to KD loss to Appendix C.
>
> > Section 6.2.3: KD Loss - “jth teacher model layer corresponds to ith student model layer” - How do we get this correspondence?
>
> For our fixed teacher model, BERT-base, which comprises 12 layers, a one-to-one sequential correspondence exists between the layers of the student and teacher models when both models have 12 layers. However, in the case of a student model with only 6 layers, the correspondence remains one-to-one, but with a 2-layer interval. This implies that the first layer of the student model corresponds to the second layer of the teacher model, and so forth, until the sixth layer of the student model aligns with the twelfth layer of the teacher model.
>
> We have incorporated this part of the content into the explanation of the formula.
>
> > Conclusion: Kindly mention some quantitative metrics here. In the current version it looks like a paraphrase of the abstract.
>
> We have revised the Conclusion section according to your suggestions.

---

> > ### Author Response · Authors · 2023-11-23
> > **Response to Reviewer pvZH (2/2)**
> >
> > > Table 5 results are close to each other. Thereby showing that the weight-weighted version might not improve the results too much as compared to Vanilla. A deeper qualitative analysis is required to establish the usage which is missing in the current version of this work.
> >
> > In fact, due to the nature of the search space, any zero-shot proxy has a baseline score. We have applied the zero-shot proxy mentioned in the related work to the search space illustrated in Figure 3. The relevant results have been added to Tables 2 and 3, and the discussion in Section 6.3 has been expanded.

---

### Official Review · Reviewer_8KoX · 2023-11-02

**Soundness:** 3 good
**Presentation:** 2 fair
**Contribution:** 2 fair
**Rating:** 3
**Confidence:** 4

**Summary:**

As the model size of large language models continues to increase, the development of lightweight language models becomes increasingly significant. While neural architecture search (NAS) is commonly used for this purpose, it often encounters biased metrics and inefficiencies. This paper introduces two evaluation proxies, specifically parameter count and principal component analysis (PCA) value, which eliminate the need for gradients and enhance efficiency. Experiments conducted on GLUE and SQuAD demonstrate the effectiveness of this approach.

**Strengths:**

1. The proposed algorithm significantly enhances the search efficiency of NAS.
2. The models discovered through this method outperform other baseline methods in GLUE and SQuAD benchmarks.
3. The visualization of the correlation between principal component analysis (PCA) and the number of parameters (#params) rankings helps interpret the effectiveness of the paper.

**Weaknesses:**

1. The paper primarily focuses on experiments with the BERT model, and it would be beneficial to conduct more experiments on other types of language models, such as generative models or larger-sized models, to ascertain the method's applicability.
2. A more in-depth discussion of the proposed method is needed. It would be beneficial to conduct ablation studies to assess the impact of each loss term, considering that multiple loss terms are included in the fine-tuning process.
3. The differences in training datasets and objectives compared to previous works make it unclear whether the improvement stems from the searched architecture or other factors.
4. The improvement observed in GLUE and SQuAD is described as marginal.
5. This article requires significant improvement in writing and formatting. The misplaced tables contribute to a lack of clarity in the paper's presentation, such as in section 6.3.

**Questions:**

1. Could you provide information regarding the zero-shot performance of the searched model? Additionally, I'm interested in learning about the performance of architectures discovered using W-PCA with different parameters.
2. Have you conducted experiments with multiple shots and employed an iterative searching strategy for multiple shots searching?

---

> ### Author Response · Authors · 2023-11-23
> **Response to Reviewer 8KoX (1/2)**
>
> Thank you for your valuable feedback and constructive comments have been instrumental in shaping our research.
>
> >The paper primarily focuses on experiments with the BERT model, and it would be beneficial to conduct more experiments on other types of language models, such as generative models or larger-sized models, to ascertain the method's applicability.
>
> We conducted experiments using a larger-sized model by increasing the size of the search space, as described in Section 6.2.1. Specifically, we doubled the hidden\_size and the hidden\_dimension of $n$ candidate dimensions. Additionally, we raised the parameter limit in the genetic algorithm to 67M, resulting in our W-PCA-Large model. As shown in the Table, despite having a slightly lower parameter count, our model outperforms TinyBERT-6 [1] and EfficientBERT [2] models of similar scale in terms of average GLUE score. This indicates that our proposed zero-shot proxy also demonstrates good adaptability in larger search spaces.
>
> |  Model   | #Params  |  QNLI   | MRPC  |  SST-2   | CoLA |  STS-B   | MNLI-m/mm  | RTE  |  QQP   | AVG  |
> |  ----  | ----  |  ----  | ----  |  ----  | ----  |  ----  | ----  | ----  |  ----  | ----  |
> | TinyBERT-6 [1]  |  67.0M  | 89.8 | 89.0 | 92.0 | 38.8 | 83.1 | 83.8/83.2 | 65.8 | 71.4 |77.4 |
> | EfficientBERT [2]  | 70.1M  | 90.4 | 89.0 | 92.6 | 46.2 | 83.7 | 84.1/83.2 | 67.7 | 74.4 | 78.7 |
> | W-PCA-Large  | 66.9M  | 90.9 | 88.7 | 93.0 | 40.0 | 87.5 | 84.6/83.3 | 75.6 | 71.5 | 79.5 |
>
> We have included this part of the content in Appendix E.1 of the paper (Page 13).
>
> > A more in-depth discussion of the proposed method is needed. It would be beneficial to conduct ablation studies to assess the impact of each loss term, considering that multiple loss terms are included in the fine-tuning process.
>
> We utilized the components of W-PCA, as described in Equation (4) where the first component is the number of parameters (\#Params) and the second component is the V-PCA value (defined in Equation (2)), as fitness values for the genetic algorithm to explore the optimal network structure in Section 6.2.1. We then compared the performance of the discovered network structures with W-PCA.
>
> The results, presented in Table 5, demonstrate that by multiplying the number of parameters with the V-PCA value and using W-PCA as the zero-shot evaluation metric, the performance of the searched networks significantly improves compared to using either \#Params or V-PCA alone as the evaluation metric.
>
> We have added this explanation to Section 6.5 of the original text.
>
> > The differences in training datasets and objectives compared to previous works make it unclear whether the improvement stems from the searched architecture or other factors.
>
> We have applied the zero-shot proxy from related work to the search space we designed (Figure 3), and the results from GLUE test set are as follows.
>
> |  Model   |   Type | #Params   |  Time  |  QNLI   | MRPC  |  SST-2   | CoLA |  STS-B   | MNLI-m/mm  | RTE  |  QQP   | AVG  |
> |  ----  | ----  | ----  | ----  | ----  | ----  |  ----  | ----  |  ----  | ----  | ----  |  ----  | ----  |
> | Synaptic Saliency [3] |zero-shot |15.7M | 58ms | 89.4 | 88.1 | 91.0| 33.6 | 83.1 | 82.6/81.1 | 70.6|  70.3 | 76.6 |
> | Activation Distance [4] | zero-shot |15.6M | 60ms | 88.9 | 87.6 | 91.2 | 30.7 | 82.9 | 81.1/80.4 | 70.4| 70.1 |75.9 |
> | Synaptic Diversity [5] | zero-shot |15.6M | 57ms | 88.3 | 88.1 | 91.5 | 25.8 | 84.7 | 81.3/80.2 | 70.6| 70.3 |75.6  |
> | Head Confidence [6] |  zero-shot |15.6M | 63ms | 89.5 | 88.3 | 92.4 | 31.7 | 85.7 | 82.8/81.9 | 74.0 | 70.9 |77.5  |
> | Softmax Confidence[6] |  zero-shot |15.6M | 61ms | 88.4 | 87.5 | 90.8 | 32.5 | 83.5 | 81.2/80.5 | 70.3| 69.9 |76.1  |
>
> The results from GLUE dev set are as follows.
>
> |  Model   |   #Params  |  Time  |  QNLI   | MRPC  |  SST-2   | CoLA |  STS-B   | MNLI-m/mm  | RTE  |  QQP   | AVG  |
> |  ----  | ----  | ----  | ----  | ----  |  ----  | ----  |  ----  | ----  | ----  |  ----  | ----  |
> | Synaptic Diversity [5] | 15.6M | 0.7 d | 88.9 | 87.6 | 91.4 | 32.0 | 84.1 | 81.0 | 73.4| 88.2 |78.3  |
> | Head Confidence [6] |  15.6M | 0.5 d | 90.1 | 89.7 | 92.4 | 37.5 | 84.1 | 82.5 | 75.9| 89.1 |80.2  |
> | Softmax Confidence[6] |  15.6M | 0.5 d | 89.4 | 88.3 | 92.0 | 32.6 | 84.7 | 81.6 | 73.9| 88.9 |78.9  |
>
> We have added the comparative results of this part in the same search space to Table 2 and Table 3, and discussed the relevant content in the main text.
>
> >  The improvement observed in GLUE and SQuAD is described as marginal.
>
> We have expanded the relevant discussions in Sections 6.3 and 6.4.
>
> > This article requires significant improvement in writing and formatting. The misplaced tables contribute to a lack of clarity in the paper's presentation, such as in section 6.3.
>
> We have reorganized the relevant tables and improved multiple expressions in the original text. You can also refer to the revisions made based on the guidance of Reviewer pvZH.

---

> ### Author Response · Authors · 2023-11-23
> **Response to Reviewer 8KoX (2/2)**
>
> > Could you provide information regarding the zero-shot performance of the searched model? Additionally, I'm interested in learning about the performance of architectures discovered using W-PCA with different parameters.
>
> We conducted supplementary experiments on different initialization parameters, which can be seen in the newly added Appendix B. Additionally, the discussion on model visualization has been included in Appendix D.
>
> > Have you conducted experiments with multiple shots and employed an iterative searching strategy for multiple shots searching?
>
> There is limited research on n-shots (n > 1) NAS in NLP tasks. The only one we are aware of is [7], as it requires significant computational resources. We have conducted supplementary experiments for the n = 1 case, and the results are as follows:
>
> |  Model   |   Type | #Params   |  Time  |  QNLI   | MRPC  |  SST-2   | CoLA |  STS-B   | MNLI-m/mm  | RTE  |  QQP   | AVG  |
> |  ----  | ----  | ----  | ----  | ----  | ----  |  ----  | ----  |  ----  | ----  | ----  |  ----  | ----  |
> | W-PCA-Tiny |zero-shot | 9.6M | 0.4 d | 88.7 | 87.6 | 91.9 | 27.4 | 84.8 | 81.1/79.8 | 71.1| 70.3 |75.9 |
> |  | one-shot |9.7M | 24 d | 89.2 | 87.5 | 92.3 | 28.9 | 83.7 | 81.4/80.5 | 71.4 | 70.5 |76.2 |
> | W-PCA-Small | zero-shot |15.6M | 0.5 d | 90.3 | 88.7 | 91.5 | 38.4 | 86.4 | 82.8/82.2 | 73.8 | 70.8 |78.3  |
> | |  one-shot |15.6M | 28 d | 90.3 | 88.9 | 92.5 | 36.1 | 86.7 | 83.7/82.5 | 74.4 | 70.6 |78.4 |
>
> Despite investing a significant number of GPU days in the one-shot NAS search, the performance improvement on various-sized models of W-PCA is not significant. Zero-shot NAS remains the most cost-effective search solution.
>
> The relevant content has been included in Appendix E.2 （Page 14）
>
> [1] Jiao et al.Tinybert: Distilling bert for natural language understanding.  EMNLP 2020.
>
> [2] Dong et al. Efficientbert: Progressively searching multilayer perceptron via warm-up knowledge distillation.  EMNLP 2021.
>
> [3] Abdelfattah et al. Zero-costproxies for lightweight nas. ICLR 2020.
>
> [4] Mellor et al. Neural architecture search withouttraining. ICML 2021.
>
> [5] Zhou et al.  Training-free transformer architecture search. CVPR 2022.
>
> [6] Aaron Serianni and Jugal Kalita. Training-free neural architecture search for RNNs and transform-
> ers. ACL 2023
>
> [7] So D, Le Q, Liang C. The evolved transformer. ICML 2019.

---

### Author Response · Authors · 2023-11-23
**Summary of Paper Revision**

We have made the following modifications to the paper based on the reviewer's suggestions:

1. Added detailed descriptions of the training details of the space of FlexiBERT in Appendix A.
2. Included a discussion on the ranking correlation of different initialization parameters in Appendix B.
3. Moved the description of KD loss from the original Section 2.3 to Appendix C, and added descriptions of teacher and student distillation for corresponding layers.
4. Added a discussion on model visualization in Appendix D.
5. Included a discussion on slightly larger models and one-shot NAS in Appendix E.
6. Added a discussion on other zero-shot proxies in the search space in Table 2 and Table 3.
7. Expanded many parts of the original text and addressed some formatting issues.

---

### Meta-Review · Area_Chair_aXnh · 2023-12-05

**Metareview:**

This paper introduces a zero-shot Neural Architecture Search (NAS) method using evaluation proxies like parameter count and PCA values, aiming to expedite the selection of lightweight language models. Reviewers highlight the method's potential for enhancing NAS efficiency and achieving competitive results on GLUE and SQuAD benchmarks. Strengths include the significant reduction in training time, detailed implementation description aiding reproducibility, and the novel application of 0-shot NAS to NLU tasks. However, concerns are raised about limited model applicability beyond BERT, lack of clarity in presenting results, and the method's marginal performance improvements. Reviewers suggest deeper qualitative analyses, ablation studies, and discussions on trade-offs between model size, training time, and performance. Overall, despite promising aspects, the paper needs substantial revisions addressing clarity, broader applicability, and deeper analyses to be accepted.

**Justification For Why Not Higher Score:**

Despite improvements in efficiency, the paper lacks comprehensive architectural guidelines and universally applicable designs. While some experiments showed promise, questions remain regarding the method's scalability and its effectiveness in resource-constrained settings.

**Justification For Why Not Lower Score:**

N/A

---

### Decision · Program_Chairs · 2024-01-16

Reject